# Male rat leukocyte population dynamics predict a window for intervention in aging

Hagai Yanai[1]*[†], Christopher Dunn[2][†], Bongsoo Park[1], Christopher Coletta[3], Ross A McDevitt[4], Taylor McNeely[1], Michael Leone[1], Robert P Wersto[2], Kathy A Perdue[4], Isabel Beerman[1]*

[1]Epigenetics and Stem Cell Unit, Translational Gerontology Branch, National Institute on Aging, Baltimore, United States; [2]Flow Cytometry Core, National Institute on Aging, Baltimore, United States; [3]Computational Biology and Genomics Core, Laboratory of Genetics & Genomics, National Institute on Aging, Baltimore, United States; [4]Comparative Medicine Section, National Institute on Aging, Baltimore, United States

**Abstract** Many age-associated changes in the human hematopoietic system have been reproduced in murine models; however, such changes have not been as robustly explored in rats despite the fact these larger rodents are more physiologically similar to humans. We examined peripheral blood of male F344 rats ranging from 3 to 27 months of age and found significant age-associated changes with distinct leukocyte population shifts. We report CD25+ CD4+ population frequency is a strong predictor of healthy aging, generate a model using blood parameters, and find rats with blood profiles that diverge from chronologic age indicate debility; thus, assessments of blood composition may be useful for non-lethal disease profiling or as a surrogate measure for efficacy of aging interventions. Importantly, blood parameters and DNA methylation alterations, defined distinct juncture points during aging, supporting a non-linear aging process. Our results suggest these inflection points are important considerations for aging interventions. Overall, we present rat blood aging metrics that can serve as a resource to evaluate health and the effects of interventions in a model system physiologically more reflective of humans.

**\*For correspondence:**
hagai.yanai@nih.gov (HY);
isabel.beerman@nih.gov (IB)

[†]These authors contributed equally to this work

**Competing interest:** The authors declare that no competing interests exist.

## Editor's evaluation

This paper uses flow cytometry to characterize the changes in immune cell composition of the peripheral blood, as well as DNA methylation, during aging in male rats. Using this data, the authors were able to observe distinct cell composition and DNA methylation profiles with age, providing predictive measures of aging. Additionally, the authors identified a novel marker, CD25+ T cell frequency, as a predictor of age in rats. This resource will be useful to the community as a well-controlled dataset dissecting changes in circulating blood cells with aging in an important mammalian model.

## Introduction

Among all mammalian tissues, blood is perhaps the easiest to collect in relatively large quantities for various advanced analyses, with modest discomfort to the donor. This has made blood the prevailing tissue for a wide variety of applications requiring a sizable amount of material, such as: clinical

**eLife digest** Our blood contains many types of white blood cells, which play important roles in defending the body against infections and other threats to our health. The number of these cells changes with age, and this in turn contributes to many other alterations that happen in the body as we get older. For example, the immune system generally gets weaker at fighting infections and preventing other cells from developing into cancer. On top of that, the white blood cells themselves can become cancerous, resulting in several types of blood cancer that are more likely to happen in older people.

Many previous studies have examined how the number of white blood cells changes with age in humans and mice. However, our understanding of this process in rats is still poor, despite the fact that the way the human body works has more in common with the rat body than the mouse body.

Here, Yanai, Dunn et al. have studied samples of blood from rats between three to 27 months old. The experiments found that it is possible to accurately predict the age of healthy rats by measuring the frequency of populations of white blood cells, especially a certain type known as CD25+ CD4+ cells. If the animals had any form of illness, their predicted age deviated from their actual age. Furthermore, while some changes in the blood were gradual and continuous, others displayed distinct shifts when the rats reached specific ages.

In the future, these findings may be used as a tool to help researchers diagnose illnesses in rats before the animals develop symptoms, or to more easily establish if a treatment is having a positive effect on the rats' health. The work of Yanai, Dunn et al. also provides new insights into aging that could potentially aid the design of new screening methods to predict cancer and intervene using a model system that is more similar to humans.

diagnosis, immune surveillance, and molecular investigations, including DNA methylation (*Fahy et al., 2019*; *Levine et al., 2020*).

To date, peripheral blood aging has been largely characterized in humans (*Márquez et al., 2020*; *Mahlknecht and Kaiser, 2010*; *Wayne et al., 1990*) and mice (*Pinchuk and Filipov, 2008*). Studies in mouse models have provided important insights as this model system allows investigation of the impact of a variety of genetic interventions and hematopoietic stem cell (HSC) transplant assays. Many murine aging phenotypes accurately reflect reports from human studies including a decline in immunogenic response of lymphoid cells (*Pinchuk and Filipov, 2008*), decreased CD4/CD8 T cell ratio (*Nikolich-Zugich, 2008*), and an increase in myeloid cells at the expense of lymphoid cells (*Pang et al., 2011*). These phenotypes have been shown to be influenced, at least in part, by intrinsic HSC aging (*Beerman et al., 2010*).

However, the mouse has drawbacks as a model of human hematopoiesis including the fact that the common laboratory strains do not reliably develop spontaneous blood cancers, which is a common aging phenotype in humans (*Zjablovskaja and Florian, 2019*). The differences between the two species are further exemplified by the fact that mouse blood is dominated by lymphocytes whereas human blood is rich in neutrophils (*Mestas and Hughes, 2004*); which in the murine model may dampen the impact of the age-associated decrease in lymphoid potential.

An attractive alternative system is the rat model. Rat physiology is more analogous to humans (*Blais et al., 2017*; *Gibbs et al., 2004*), yet these rodents still have relatively short lifespans (*Turturro et al., 1999*) in which to study aging phenotypes. Furthermore, the rat and human genomes share genes involved in immunity and hematopoiesis absent in mice (*Gibbs et al., 2004*). Importantly, rats display age-related incidence of leukemia, especially the Fischer 344 CDF (F344) strain (*Ward and Reynolds, 1983*). All the above suggest the rat may be better suited to study the aging of the blood and immune systems. Despite these advantages, use of the rat model is less common for hematopoietic studies as mice are smaller, generally cost less to maintain, and transgenic tools to manipulate the murine genomes have been extensively developed. As a result, the impact of aging on the blood compartment in rats is not well characterized.

We sought to describe changes in the composition of the peripheral blood during aging in the F344 male rat using flow cytometry. Additionally, we found that CD25+ T cell frequency is a novel marker for predicting aging, and rats have defined age-associated inflection points linked to altered

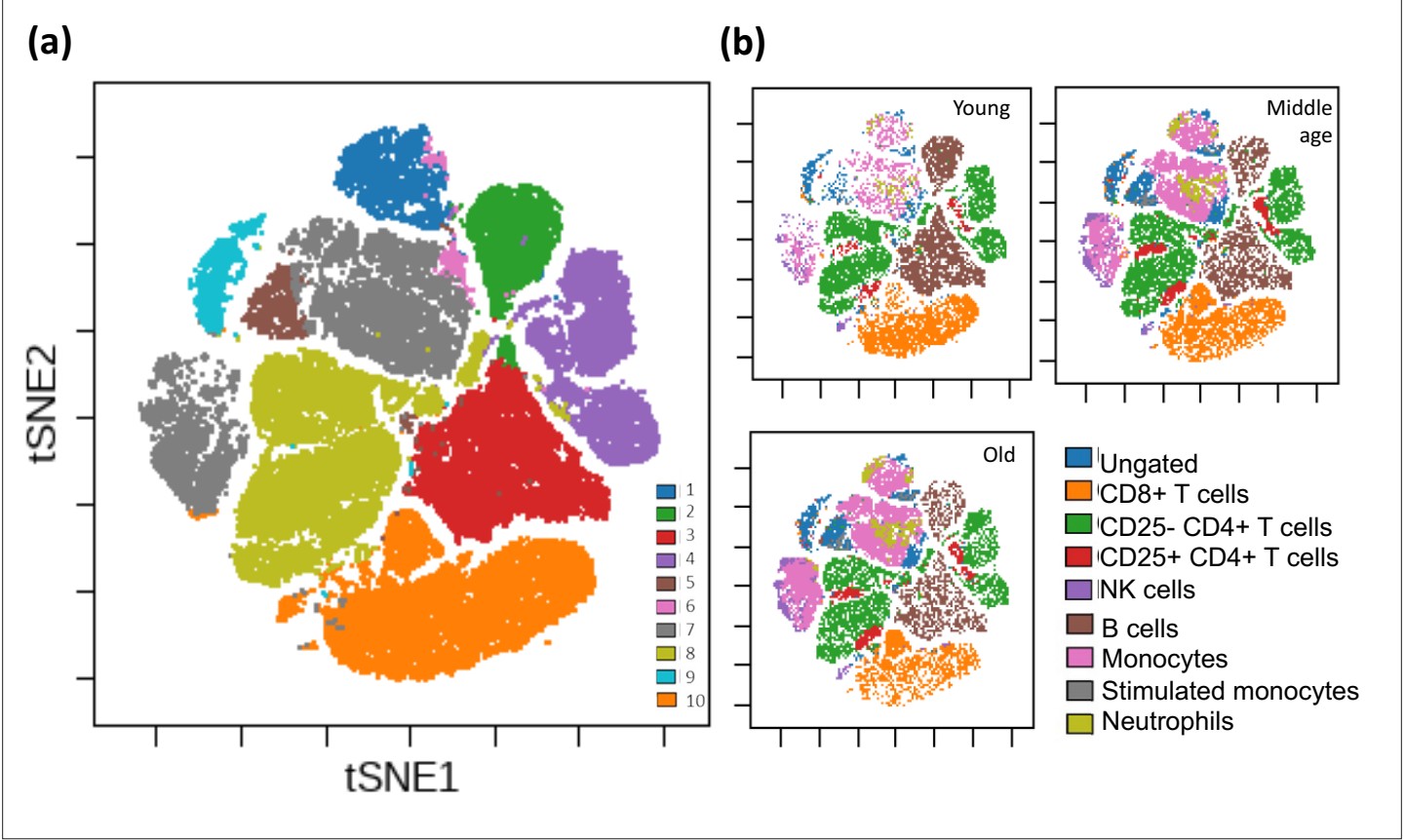

**Figure 1.** Unsupervised clustering of all F344 rat peripheral blood cells. Clustering of rat peripheral blood using a self-organizing map algorithm. For detailed clustering also see *Figure 1—figure supplement 1*. (**a**) Overall clustering of all leukocytes from 146 rats overlaid on a t-SNE map; each cluster is denoted as a different color (see legend). (**b**) Specific blood cell types illustrated on the same t-SNE map for three age groups (n = 10,000 cells from each age group; total 30,000 cells). For visualization purposes, rats were divided into age groups: young (3–4 months), middle-aged (15–23 months), and old (24+ months).

The online version of this article includes the following figure supplement(s) for figure 1:

**Figure supplement 1.** Self-organizing map clustering of F344 rat leukocytes.

**Figure supplement 2.** Gating strategy.

methylation profiles and cell composition. These results indicate that age-associated blood pheno-types in rats provide relevant insight into human blood aging and point to a critical time period of hematopoiesis which may best be targeted for anti-aging interventions.

## Results

### Aging patterns in rat peripheral blood leukocyte populations

We analyzed the peripheral blood composition of 146 male F344 rats ranging in age from 3 to 27 months, using a panel of seven monoclonal antibodies to evaluate major leukocyte populations. As cell-specific gating strategies for rat blood populations are less defined than in humans or mice, we initially evaluated the flow cytometry results in an unbiased manner using t-distributed stochastic neighbor embedding (t-SNE) and self-organizing maps analyses. Cells generally clustered in a manner that allowed for identification of defined populations (*Figure 1a* and *Figure 1—figure supplement 1*) and clustering analysis helped inform decisions for gating strategies we devised for cell-type classification (*Figure 1—figure supplement 2*). Age-related changes were also profiled from single-cell analyses and revealed an accumulation of cells in clusters corresponding to myeloid lineage cells and a reduction of cells in clusters expressing markers associated with B and CD8[+] T cells (*Figure 1b*). Globally CD4[+] T cells decreased with age, but we identified two distinct CD4[+] populations with different

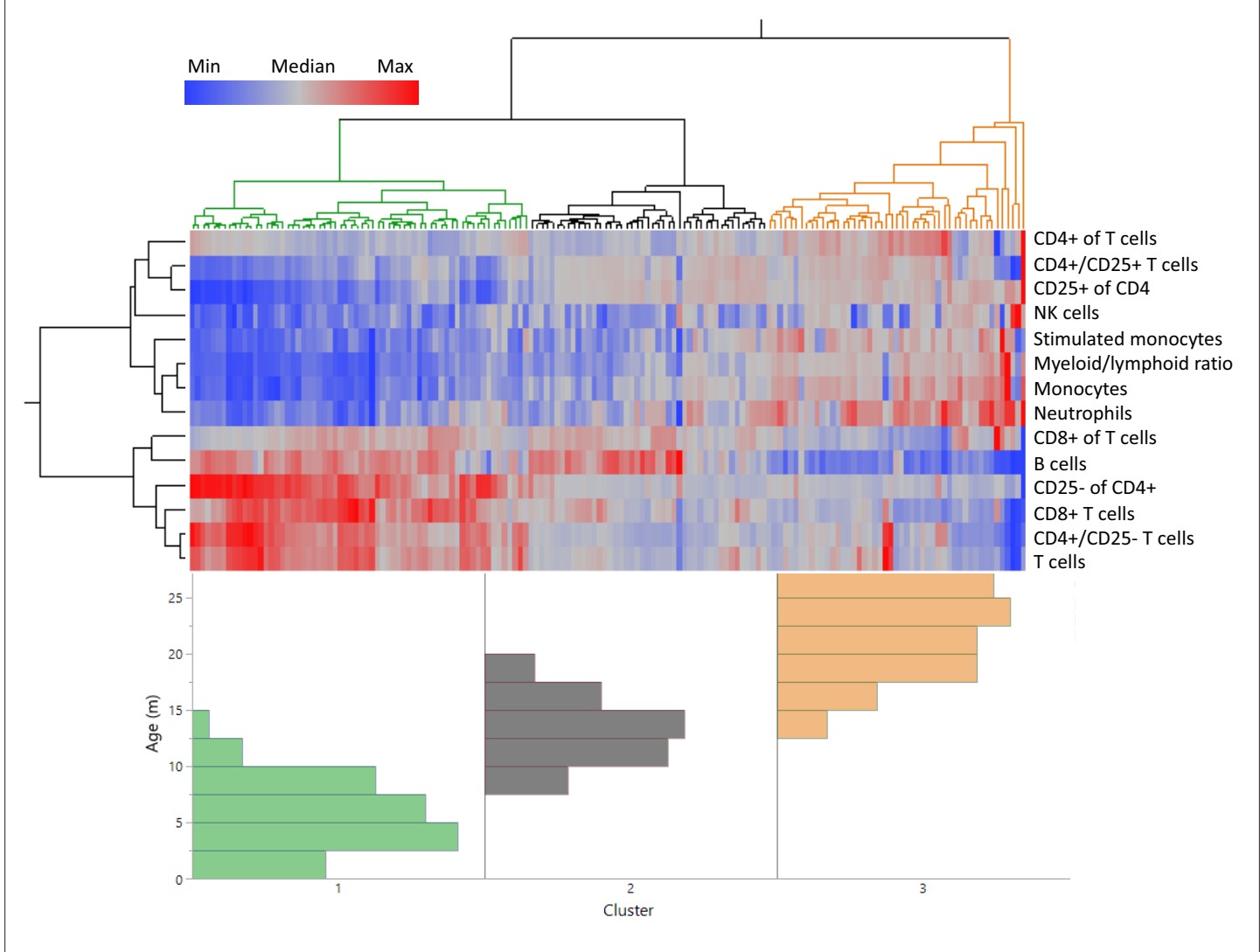

**Figure 2.** Distinct age clustering of rats based on white blood cell frequencies. Hierarchical agglomerative clustering with distance matrix by Ward's method. The top lines represent the distance matrix for the individual rats, and the three major branches are annotated by green, dark gray, and orange. The heatmap represents the normalized frequency of each population as indicated by the legend. The lines on the left indicate the distance matrix of the frequencies. The bottom histogram represents the age distribution for each of the major clusters in the same colors as above.

trajectories during aging. The CD4+ T cell cluster expressing high levels of CD25 did not show an age-associated decline in cell number and instead increased in frequency over time. t-SNE-defined clusters 5 and 9 (brown and cyan, *Figure 1a*) also show age-associated increases in frequency (same top left location as *Figure 1a*, but blue in *Figure 1b*); however, these clusters could not be fully identified from this combination of antibodies (*Figure 1—figure supplement 2*). Cluster 5, due to a high expression of CD11b/c, is most likely a type of myeloid cell, whereas cluster 9 most likely consists of debris (*Figure 1—figure supplement 1*) as it stained equally positive for all antibodies. Most identified cell populations that were categorized encompassed two or more clusters, indicating potential subpopulations we were unable to define with the panel of antibodies in this study.

We next sought to determine if age-specific blood parameter characteristics could be defined using frequencies of the distinct cell populations identified (*Figure 1—figure supplement 2*). Using unsupervised clustering of all rats based on leukocyte population frequencies, animals clustered into three discrete groups (*Figure 2*), indicating the overall fingerprint of blood composition is age dependent. The cluster classified as 'young' is characterized by a high proportion of lymphocytes in general, and T cells in particular, but with a low frequency of CD25+ CD4+ T cells.

In contrast, the 'old' cluster, on the far right, is characterized by a low overall lymphocyte frequency, and elevated proportions of both myeloid and CD25+ CD4+ T cells (*Figure 2*). This increase in CD25+ CD4+ T cells is similar to what is seen in human peripheral blood aging (*Gregg et al., 2005*), while in the murine model CD25+ CD4+ T cells only accumulate in secondary lymphoid organs (spleen and lymph nodes) but not in the peripheral blood (*Chiu et al., 2007*). This analysis also defines a specific 'middle age' profile that is distinct from both young and old phenotypes, but clusters more closely to the young rat profile than the old.

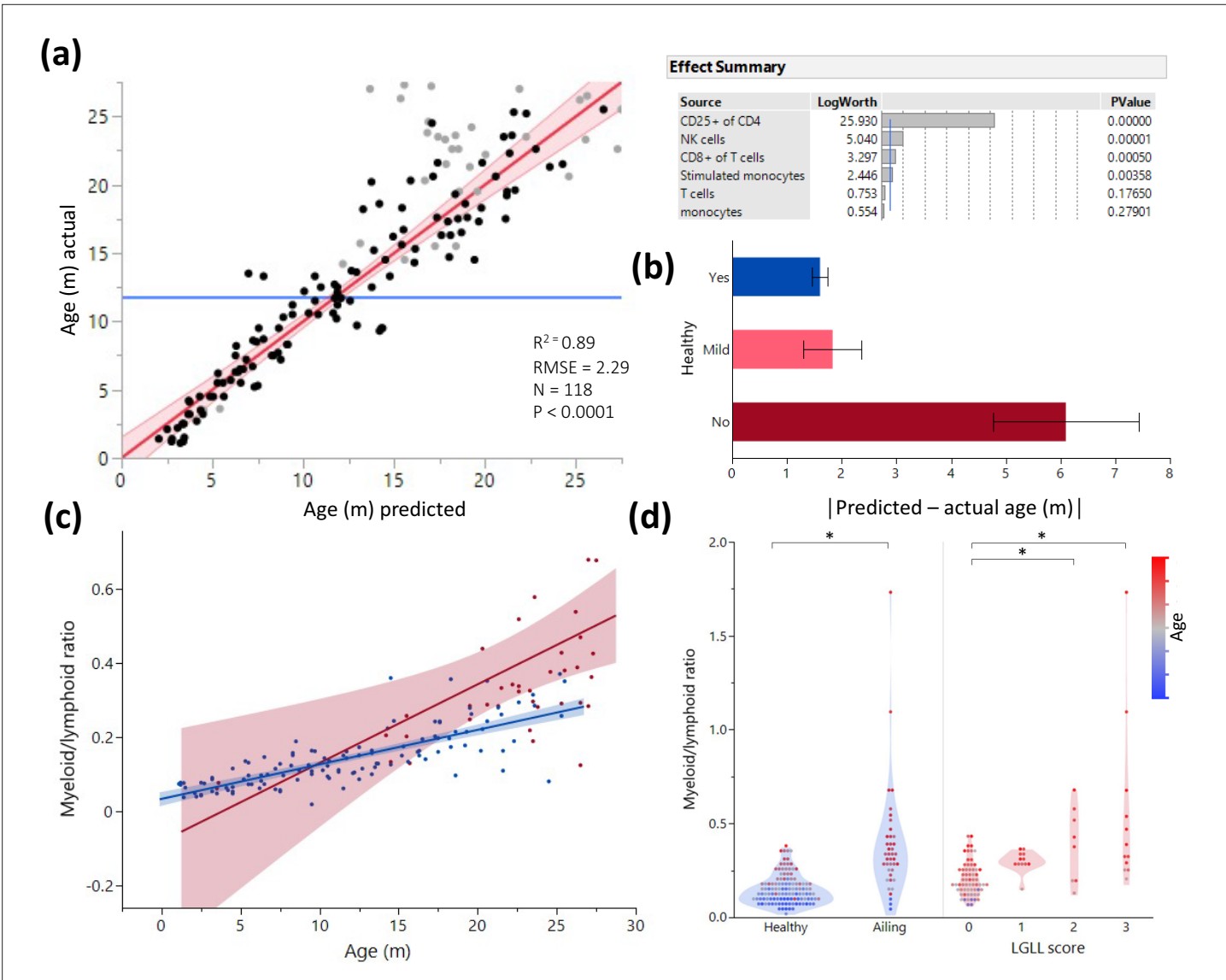

**Figure 3.** Age and illness prediction by blood leukocyte composition. All blood cell population frequencies were used to create a standard least square model of age prediction. Variables that did not contribute to the model were removed in a backstep fashion. (**a**) Predicted age vs. actual age, the red line indicating the mean and fit. Animals excluded due to illness are depicted in light gray. The effect summary for the variables used is depicted in the right panel (ffect Summary) with the Logworth cutoff depicted by a blue line. (**b**) Residual age (i.e. the difference between actual age and predicted age) plotted as absolute mean values against assessment of health as determined by necropsy. Yes = no health issues; Mild = mild health problems (e.g. minor foot lesions); No = clear health issues found during necropsy (*Source data 1*). (**c**) The correlation between myeloid/lymphoid ratio and age in healthy (blue) and ill (red) animals. (**d**) Myeloid/lymphoid ratio in the peripheral blood plotted against health status (left panel) and large granular lymphocytic leukemia (LGLL) pathology score as detected in the liver and spleen. Each dot represents a single rat. Dot color denotes age.

The online version of this article includes the following figure supplement(s) for figure 3:

**Figure supplement 1.** Prediction of pathology.

**Figure supplement 2.** Fresh vs fixed samples.

## Peripheral blood parameters predict age and pathology in rats

To assess the blood cell types most predictive of aging, we fit all leukocyte frequencies in a standard least square model to predict aging in animals with no detected pathology and then eliminated the least significant parameters in a backstep fashion (*Figure 3a*, *Figure 3—figure supplement 1*). The health status of all rats was recorded throughout the study, and necropsy was performed on all animals, less than a month post blood collection (for full health/pathology report see *Source data 1*). This enabled testing of how the health status of each rat fit into the model. One caveat was that moribund animals were excluded and euthanized per vet recommendation, so the health analysis captures mostly underlying illnesses only discernable during necropsy. The resulting blood parameter model predicted age with an approximate 2-month margin of error (as indicated by the root mean square error value). The single most predictive parameter of rat age was the proportion of CD25 expression in CD4$^+$ T cells (*Figure 3a*, right panel). Animals with histological indication of disease, including indications of large granular lymphocytic leukemia (LGLL) in spleen or liver, described in *Finesso et al., 2021*, are generally predicted by our model as more variable than their actual age (i.e. have stronger residual age values) (*Figure 3b*). When constructing an age prediction model only for ill animals, we found the myeloid/lymphoid ratio was the strongest contributor to age prediction (*Figure 3d*). Additionally, we constructed a nominal logistic regression model to predict illness and found that myeloid/lymphoid ratio is the strongest contributor, even more so than age (*Figure 3—figure supplement 1*). While the myeloid/lymphoid ratio increases with aging, this phenomenon is more pronounced in sick animals. However, this ratio alone is not sufficient to predict illness (*Figure 3d*, left) and must be combined with the other blood parameters that are characteristic of age and how variable the predicted age is from the chronologic age (*Figure 3b and c*). We also observed that myeloid/lymphoid ratio increases concurrently with severity of LGLL (*Finesso et al., 2021*; *Figure 3d*, right).

As blood samples from the 146 rats were analyzed post-fixation (see Methods), we were interested to determine whether the model generated is also applicable for freshly isolated blood. We thus generated data for both fresh and fixed rat blood, taken from the same donors. We found only small differences (<10%) in population frequencies associated with fixation, with changes in the T cell and B cell population frequencies post-fixation showing the biggest variation (*Figure 3—figure supplement 1a*). Despite these differences, applying the age-prediction model, generated from fixed blood samples, on the blood profiles generated on unfixed samples of varying ages (*Figure 3—figure supplement 1b*) demonstrated that the existing model (from fixed blood) could be used in a similar fashion to predict age with only minor mathematical adjustments.

As this study was primarily focused on male rats, we wanted to evaluate if the model could be used in a sex-independent manner. We collected a small dataset of 5 young (5 months), 5 middle-aged (15 months), and 5 old (23 months) females and compared the blood parameters to the males of the same age. We observed at the three age groups evaluated, females present sex-specific differences in leukocyte populations with notably lowered B cell and T cell percentages, and a higher frequency of neutrophils (*Figure 3—figure supplement 1c*). Interestingly, we also observed sex-specific aging trends that exacerbated the T cell and neutrophil differences. Surprisingly, despite the sex-specific phenotypes, the male generated model still robustly predicted female age (p<0.0001) (*Figure 3—figure supplement 1d*).

## Regression analysis reveals potential aging intervention points

This study involved a minimum of five rats from each month of age starting from 3 months to 27 months (*Source data 2*) which enabled us to cross-sectionally examine the dynamics of blood changes during aging. Overall, the changes in leukocyte frequencies of age-matched rats, reflected described changes in aging human blood, including a reduction in B cells and T cells, and an increase in monocytes, neutrophils, and CD25$^+$ CD4$^+$ T cells (*Figure 4—figure supplement 1*). However, while the ratio of CD4/CD8 in T cells initially declined, it increased after 15 months of age. To better understand the dynamics of these blood composition shifts, we performed a multiple regression analysis for each cell population and identified timepoints in which there was a major trajectory shift (*Figure 4a* and *Figure 4—figure supplement 1b*). Most of these shifts appear to converge at approximately 15 months of age (*Figure 4a and b*). A strong age-associated increase in the overall variance of the different parameters was also observed, which was especially pronounced at around 24 months of age

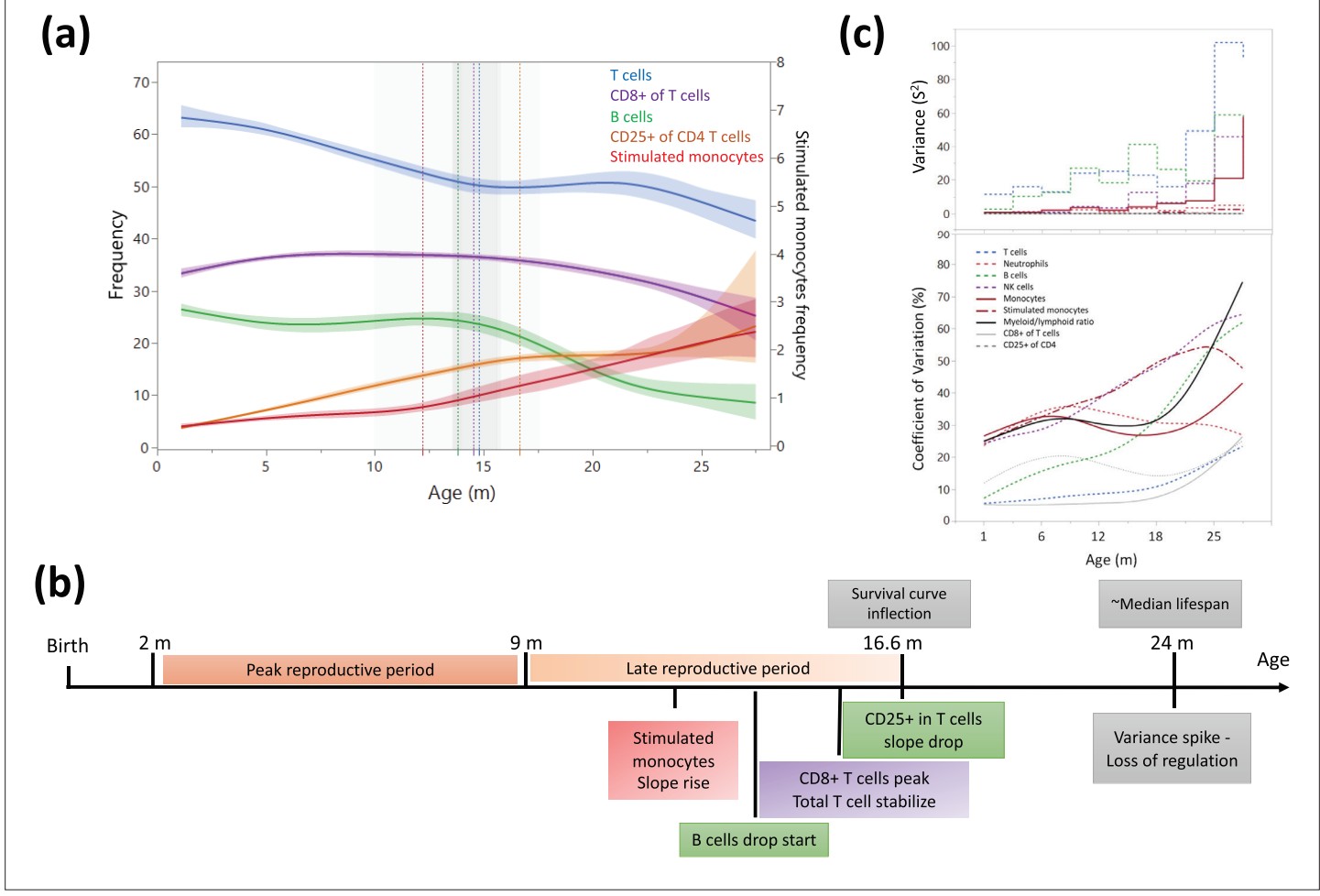

**Figure 4.** Dynamics of leukocyte frequencies during aging. (**a**) Aging trends of variables with inflection points depicted by a cubic spline regression ($\lambda$ = 0.521). Shaded colored areas indicate the fit. Inflection/breakpoints are depicted as dotted vertical lines with SE(Standard error) in shaded gray. (**b**) Summary of leukocyte critical aging points (color coded) in relation to reproductive capacity (orange) and survival curve points (gray) (***Turturro et al., 1999***). (**c**) Variance ($S^2$) and coefficient of variation for each parameter as a function of age (in a sliding window of 3 months).

The online version of this article includes the following figure supplement(s) for figure 4:

**Figure supplement 1.** Major leukocyte population frequencies as a function of age.

(***Figure 4c***). Interestingly, the first hematopoietic inflections appeared prior to the decline in survival of the classic Gompertz curve for this rat strain (***Turturro et al., 1999***).

## Peripheral rat blood DNA methylation changes with age

DNA methylation data was generated from the peripheral blood of these rats and was previously analyzed to define a rat aging 'methylation clock' (***Levine et al., 2020***). We analyzed the epigenic landscapes for hematopoietic-specific changes in methylation associated with aging using the Bismark pipeline. Global, age-dependent hypomethylation (***Figure 5a***) occurred, located mostly in intergenic and intronic regions. Promoter regions had a more balanced ratio of sites that either gained or lost methylation (***Figure 5b***). Interestingly, overall decreases in methylation were more pronounced in the transition between middle and old age compared to the global methylation changes seen in the shift from young to middle age (***Figure 5a***). To gain insight into the difference between the early-aging and late-aging changes, we performed differentially methylated region (DMR) analysis on locations near transcription start sites (TSS). Most regions with significant DNA methylation changes were progressively altered during aging, which is slightly incongruous with the phase shifts seen in blood composition. However, we also found non-linear changes in methylation using a differential methylation

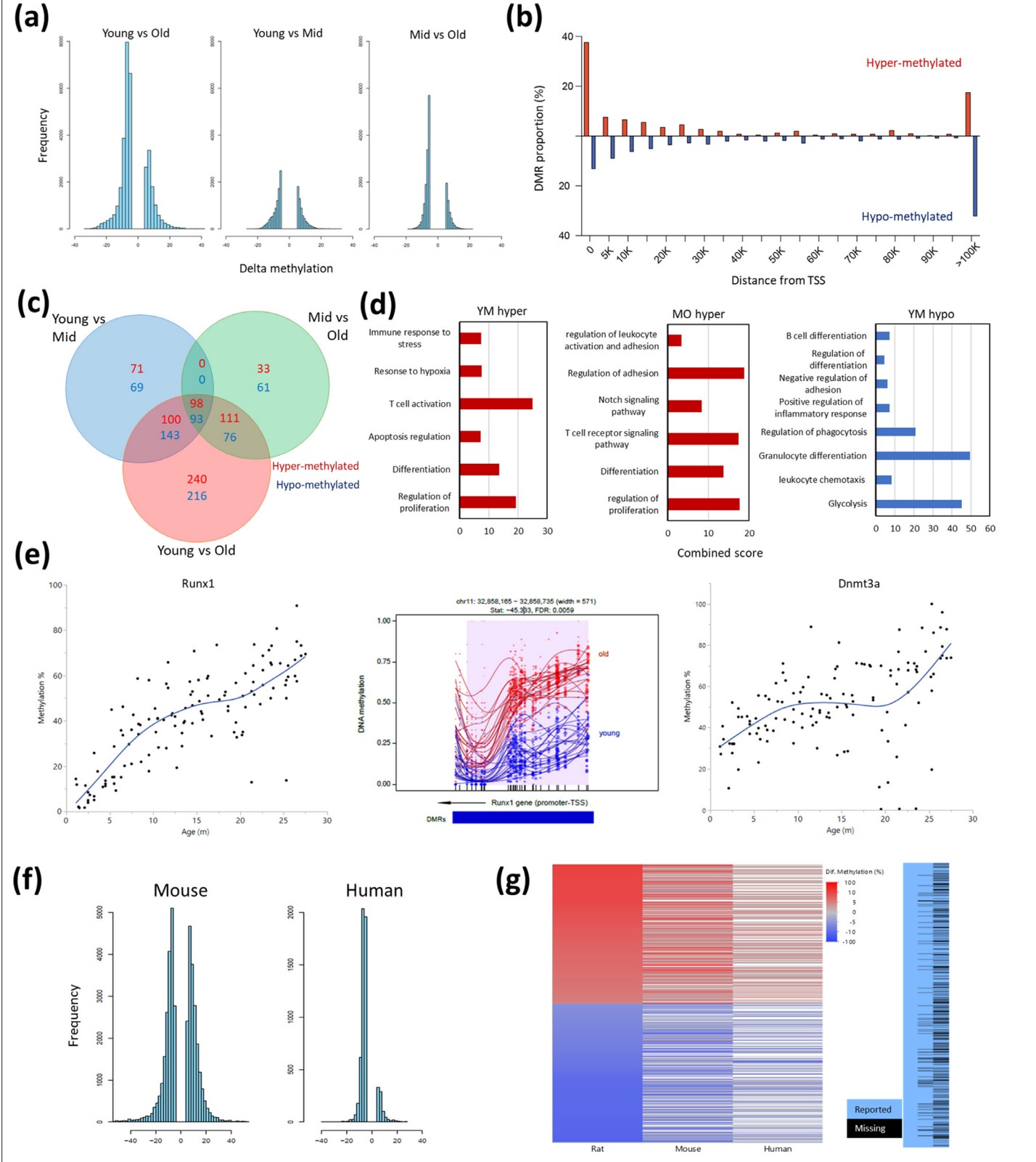

**Figure 5.** Differentially methylated regions (DMR) with rat aging. (**a**) Global differential methylation histograms per age group comparison. A cutoff of 5% methylation difference was used. (**b**) Proportion of hypermethylated and hypomethylated regions (DMR) compared to distance from the closest transcription start site (TSS). (**c**) Venn diagram for the number of DMR within 20 k bp distance of the nearest TSS based on age group comparisons. A cutoff of false discovery rate (FDR) <0.05, >5% CpG mean change, and at least 3 CpG differential DMR site per block was used. Hypermethylated

*Figure 5 continued on next page*

*Figure 5 continued*

and hypomethylated changes are indicated by the noted colors. (**d**) Summary of gene set enrichment analysis for DMRs in proximity of TSSs (<20 k bp). Analysis was performed for three age groups: young (3–8 months), middle-aged (13–17 months), and old (22–27 months). Combined score was calculated as −log10(FDR) × fold enrichment. YM - young vs. middle-aged; MO - middle-aged vs. old; YO - young vs. old. The list of hypomethylated DMRs for the MO comparison had statistically significant enrichments. For a full enrichment analysis see Table S4. (**e**) Individual methylation levels of regions near the Runx1 and Dnmt3a genes. Left panel - Runx1; middle panel - methylation map in the Runx1 TSS; right panel - Dnmt3a. (**f**) Global differential methylation histograms of mouse (*Sziráki et al., 2018*) and human (*Hannum et al., 2013*). A cutoff of 5% methylation difference was used. (**g**) Heatmap depicting concordance of DMRs near orthologous gene sites between rat (left), mouse (center), and human (right) datasets. Only genes identified as differentially methylated in old rats are shown, the rightmost panel depicts the coverage for each dataset, where missing data is black.

The online version of this article includes the following figure supplement(s) for figure 5:

**Figure supplement 1.** Response screening to determine transcription start sites that change their methylation as a function of age.

analysis of three major age groups (*Figure 5*). We identified 140 DMRs unique to the young vs. middle age comparison and 94 DMRs unique to the middle-aged vs old (*Figure 5c*). 456 DMRs appeared as significantly differential only between the young and old animals, which we attribute to a slow progressive change that only passed the cutoff of significance and fold change in the young vs old comparison (*Figure 5c*). Enrichment analysis of DMRs showed age-associated hypermethylation for several processes and included some overlaps between the early (Y-M) and late (M-O) aging phenotypes, such as regulation of T cells and differentiation (*Figure 5d*). However, some enrichments were unique to the middle age transition (regulation of apoptosis) and the late age (notch signaling pathway), again indicating a difference between the early and late aging phenotypes. Of note, while hypomethylated DMRs between young and middle age were identified for several processes (*Figure 5d*, right panel), hypomethylated DMRs in the transition from middle-aged to old were not enriched for any specific process (*Source data 3*); perhaps suggesting that these changes are more reflective of drift or random alterations and not necessarily concerted DNA methylation changes. To assess if the observed DNA methylation aging changes are specific to rats or might be more conserved, we compared our results to previously published data from mice (*Sziráki et al., 2018*) and humans (*Hannum et al., 2013*). Significant age-associated methylation changes in whole blood from both human and rat were predominantly hypomethylation; but in mice the distribution between hypomethylation and hypermethylation is more balanced (*Figure 5a and f*). When comparing the DMRs observed in rats to other species (*Figure 5g*) we find a high level of concordance, suggesting that many age-associated methylation changes may be common to several mammalian species. However, it is important to note that the sequencing was performed with different techniques for each study, resulting in a difference of coverage of orthologous loci (*Figure 5g*, right panel).

To eliminate a potential bias of predetermining age groups, we also analyzed DNA methylation at all TSS to determine if changes occurred as a function of age using the response screening platform (see Experimental Procedures) (*Figure 5—figure supplement 1a*). The most significantly changed sites were all located near the TSS of RUNX family transcription factor 1 (*Runx1*), and these sites all show strong hypermethylation with age (*Figure 5e*, left panel). The hypermethylation of *Runx1* can be mapped immediately distal to the TSS (*Figure 5e*, middle panel). The second most significantly changed DMR was located near the TSS of *Dnmt3a* (*Figure 5e*, right panel). A principal component analysis based only on the 2311 DMRs that significantly changed as a function of age (*Figure 5—figure supplement 1b*) demonstrates a marked increase in variation around the 2-year mark, analogous to the increased variation observed for population frequencies (*Figure 4c*).

## Early and late life DNA methylation switches

As the blood population frequencies indicated a mid-life inflection, we explored whether DNA methylation patterns exhibited similar signature transitions. First, we performed both linear and logistic regressions on significant age-associated DMRs near TSSs (*Figure 5—figure supplement 1a* and *Figure 6—figure supplement 1*). We next selected the regression with the best fit (based on Akaike Information Criterion, *Shavlakadze et al., 2019*). 62.9% of the sites changed linearly with age while 37.1% showed a better fit to a logistic regression (*Figure 6—figure supplement 1*). The breakpoint was calculated for sites that did not change linearly with age (*Figure 6a*), and a module enrichment analysis was performed for genes with switches before and after 15 months of age (*Figure 6b*). The methylation switch points defined converged at around 22 months of age and mildly at close to

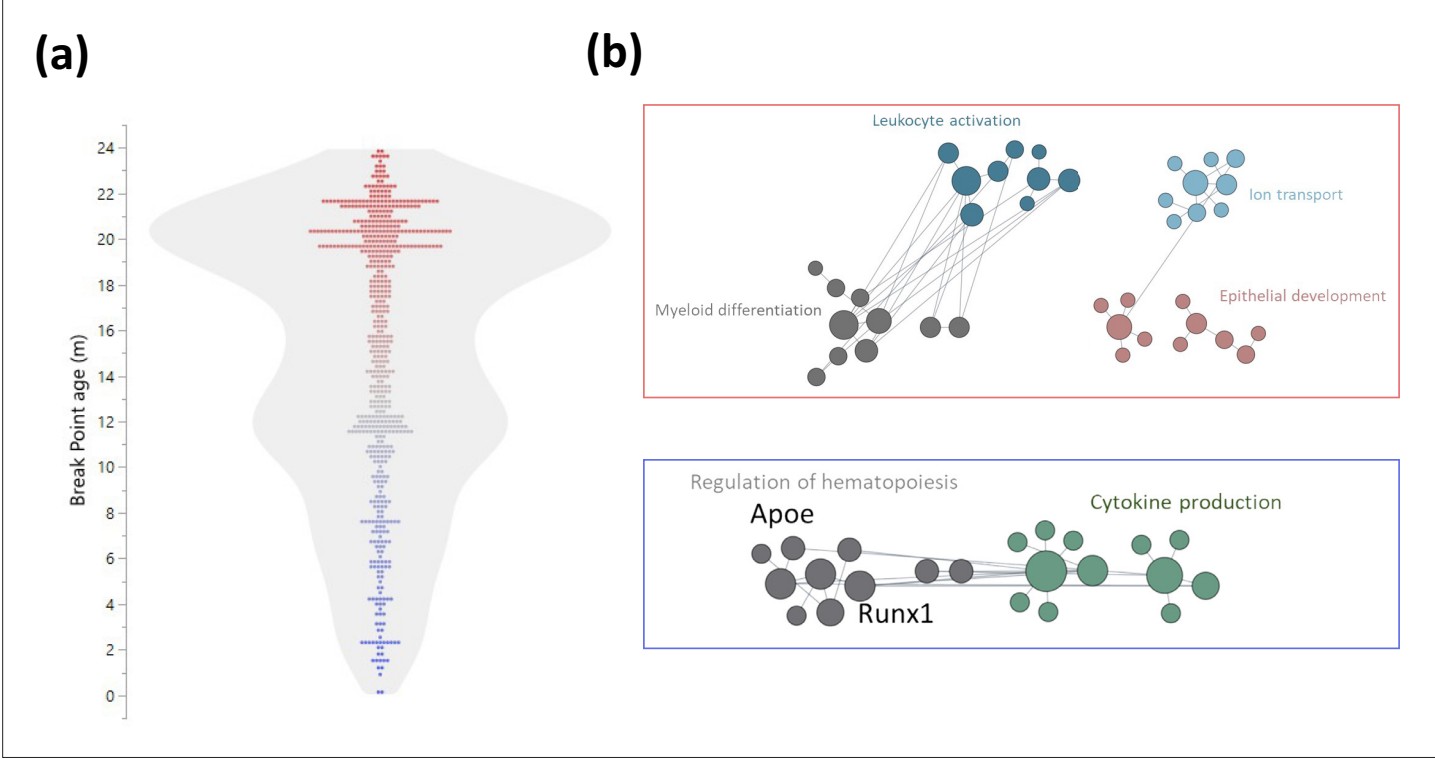

**Figure 6.** Early and late life methylation breakpoints. (**a**) Distribution of breakpoints by age. Each dot represents the breakpoint of a single gene transcription start site; color scales by age from young (blue) to old (red). Contour violin plot of the distribution is displayed in gray. (**b**) Functional module networks of the genes with breakpoints before (bottom - blue) and after (top - red) 15 months of age. Each dot represents a single gene with size corresponding to its connectivity in the network. The lines indicate interactions (edges). The networks were built for blood tissue, with human orthologs as input. The interaction network is built using the closest gene neighbors and then clustered based on enrichment in GO(Gene Ontology) categories. Networks were generated using HumanBase (https://hb.flatironinstitute.org).

The online version of this article includes the following figure supplement(s) for figure 6:

**Figure supplement 1.** Determination of methylation changes 'switch' points.

12 months of age. Genes that had a breakpoint earlier than 15 months of age were enriched for cytokine production, regulation of differentiation and hematopoiesis, and include Apolipoprotein E (*Apoe*) and *Runx1*. Genes identified to have a switch later in life were enriched for myeloid differentiation, leukocyte activation, ion transport, and epithelial development. Similar to the changes in blood parameter composition, these DNA methylation results highlight differences come in waves, notably at around 15 months, rather than occurring in a linear fashion and suggest a key remodeling timepoint of the hematopoietic system.

## Discussion

Although the rat peripheral blood compartment has been analyzed in regard to aging (*Flaherty et al., 1997*), newer panels of monoclonal antibodies offer improved insight into changes of the hematopoietic system. Here, a relatively simple blood profiling technique using eight antibodies led us to construct a model that predicts both age and, potentially, pathology. The model can be used to predict the risk of illness as a function of the deviation from predicted age to actual age, and we propose such a model could also be used to determine if intervention experiments mitigate aging phenotypes. While our dataset and analyses were for a specific set of flow cytometric parameters generated on fixed cells, we found we were able to adapt these analyses to also predict the age of freshly isolated blood (using similar flow cytometry markers). Thus, with inclusion of proper controls to account for experimental design variation, this resource of aging rat blood parameters could be adapted for studies of rat aging or pathology without requiring euthanasia of the animals. We believe this would be especially useful for longitudinal or intervention studies given the ease of blood collection at

multiple timepoints. It would also be interesting to see how our reported blood parameter changes synergize with other non-invasive measurements such as the frailty index, gait speed, etc.

This experiment was performed in male rats. In humans, clear sex aging differences have been defined, both before and after menopause (*Márquez et al., 2020*; *Gubbels Bupp, 2015*; *Huang et al., 2021*). In this study, we were surprised to observe a good fit of females to the male generated age prediction model. However, we only tested a relatively small sample of 15 females of varying ages. Additionally, we found sex-specific leukocyte frequency differences that were accentuated by age. These differences highlight potential sex-specific aging patterns that remain to be fully explored. This comparison would be especially intriguing if viewed in the context of aging of the reproductive system as in the males we observed an alignment between hematopoietic aging and the end of the late reproductive period (*Figure 4b*).

A strong predictor of age was the accumulation of CD25 on CD4$^+$ T cells in the F344 rat model. These results support previous findings that report an age-dependent accumulation of regulatory T cells (Tregs) (*Chiu et al., 2007*; *Lages et al., 2008*; *Sharma et al., 2006*). This stable increase in the F344 rat is more similar to that observed in humans (*Gregg et al., 2005*) than in mice (*Chiu et al., 2007*) and highlights the relevance of the rat model to study hematopoietic aging. *Garg et al., 2014* have previously suggested that this accumulation is driven by hypomethylation of forkhead box P3 (Foxp3) in CD25$^+$CD4$^+$ T cells; however, in our dataset analyzing total blood cell methylation, we do not see differential methylation of Foxp3. This is likely because CD25$^+$CD4$^+$ T cells are a rare subpopulation of the total blood, and DNA methylation changes in this population would not significantly affect the global blood methylation profiles.

We observed an age-dependent myeloid bias with rat aging, similar to mice (*Beerman et al., 2010*) and humans (*Pang et al., 2011*). The observed myeloid bias was a strong predictor of pathology in general, and of LGLL in particular. Importantly, strong myeloid bias was correlated with illness that was otherwise only noticeable during autopsy and undetectable during routine veterinary observation. A caveat to this observation is that myeloid bias increases with age, even in healthy animals, meaning that the myeloid/lymphoid ratio can only be useful as a diagnostic parameter when plotted against the expected 'normal' age-dependent increase. In this reference dataset, we found a significant association between severity of LGLL and increased myeloid cell numbers. The myeloid bias we report in rats has previously been documented in mice; however, most mouse models do not develop age-associated blood diseases seen in humans. Thus, we find the correlation between increased myeloid cells and LGLL severity in rats to be of particular relevance for potential modeling of human blood aging phenotypes.

Our analysis of the age-related changes in leukocyte profiles indicates inflection points in the trajectories occurring at early middle age (~15 months) and at ~24 months of age. The first inflection point is characterized by several shifts in leukocyte frequencies, including a stabilization of T cell decline, peak in CD8$^+$ T cells, rapid drop of B cells, deceleration in the accumulation of CD25$^+$ CD4$^+$ T cells, and an acceleration in the increase of stimulated monocytes. The second shift point we observed is defined by a marked increase in variance between individual rats which alludes to a pan-hematopoietic loss of homeostasis (*Figure 4c*). Alternatively, this increased variability could be the result of a divergence in biological aging rates between individual rats. It would be interesting to investigate this further in a longitudinal study that can make a clear distinction between these postulations. We hypothesize that interventions aiming at mitigating aging phenotypes would be significantly more effective if performed prior to these defined shift points. The inflection points in the blood are similar to the transition pattens (linear, early logistic, and mid-logistic) of gene expression on tissues isolated from aging Sprague-Dawley rats (*Shavlakadze et al., 2019*). Furthermore, supporting a claim for interventions predating the observed switch point are studies of caloric restriction (CR). It is quite clear that early onset CR has more benefits than late onset, but it is yet unclear what the ideal starting age should be (*Ingram and de Cabo, 2017*). In fact, late onset CR has even been shown to be potentially detrimental to cognitive function (*Todorovic et al., 2018*). However, Chen et al. (*Chen et al., 2015*) reported a beneficial effect of late onset CR to muscle mass and metabolism in Sprague-Dawley rats, indicating that the ideal points of intervention could vary greatly between different systems. In fact, interventions that aim to prevent or delay aging could be fundamentally different than those that aim to reverse it, even to the extent where harm could be caused when one strategy is applied instead of the other (*Hadley et al., 2005*). It is therefore critical to understand which systems display aging

switch points, and when, in order to design the appropriate intervention regimen. This is especially true when translating experimental paradigms to human interventions due to the great heterogeneity between the aging rates of both individuals and different systems within the individual (*Gonzalez-Freire et al., 2020*).

The combined results of changes in population frequencies and the DNA methylome indicate that hematopoietic aging of rats occurs in phases as opposed to a continuous process. However, with the current data, we cannot ascertain the drivers underlying the shift between these aging phases, whether they are triggered by a specific event, or simply manifest when a critical mass of small changes reaches a threshold. Whichever the case, investigating these relatively early timepoints in which blood profiles begin to irreversibly shift, indicating critical trajectory alterations, is of great interest.

The age-related DNA methylation changes in all leukocytes highlight the potential involvement of epigenetic changes that have occurred in HSCs and/or early progenitors, and are transmitted to all the differentiated progeny, as indicated by the hypermethylation of *Runx1* and the gene enrichment analysis. In addition, the lack of enrichment for hypomethylated TSS in the transition from middle age to old indicates that age-related gains of methylation may be a directed process, whereas loss of methylation later in life is more non-specific and perhaps attributed to drift. If so, mitigating the seemingly random loss of methylation would be an interesting potential aging intervention. Intriguingly, there is a surprising concordance between the rat and human differential methylation patterns, both in terms of a predominant global loss of methylation with age and in specific DMRs that are near orthologous genes. However, a more comprehensive coverage of overlapping methylation sites is needed to validate these initial findings. As we observed a change of leukocyte frequencies with age consistent between species, and different cell types present different DNA methylation profiles, it would be interesting to ascertain in future studies to what extent the observed methylation change rely on distinct changes in blood composition. However, we observed different breakpoints for the population frequencies and the DNA methylation, which indicate a slight disconnect between the two. This supports the notion that at least some of the methylation changes are derived from aging alterations in stem/progenitor cells that are inherited by their progeny (*Beerman et al., 2013*).

In summary, we present this work as a resource for investigators studying aging in the rat model, which appears to be a robust system for modeling the aging human hematopoietic system. Our research emphasizes the importance of studying changes in homeostasis during aging as we present key trajectory shifts in blood populations that occur at relatively early age (15 months). Finally, our research model describes aging 'juncture points'. Given these specific breakpoints in composition of the hematopoietic system, we posit that strategic aging interventions would likely be more robust and beneficial if performed earlier in life to mitigate or stall the major changes that occur under homeostatic conditions. While our data support that the male-derived model can be used for female predictions, it was evident there are sex-specific differences in the blood of rats. We hope these data will be expanded upon in a female longitudinal study, in which frailty measurements would also be included. It would be exciting to further explore if an illness prediction model based off of these defined blood parameters could be extrapolated for a broad range of diseases, and ultimately if similar age-associated shifts occur in other organs/tissue systems, animal models, and specifically in humans.

# Methods

**Key resources table**

| Reagent type (species) or resource | Designation | Source or reference | Identifiers | Additional information |
| --- | --- | --- | --- | --- |
| Strain, strain background (Rattus Norvegicus) | Fischer 344 male and female (F344-cdf) | NIA aging colony | F344-cdf | Housed at the Charles River Laboratories (Frederick, MD) |
| Antibody | (Mouse monoclonal) Anti-rat CD3-FITC, Clone 1F4 | BioLegend | Cat#201,403 | (1:200) |
| Antibody | (Mouse monoclonal) anti-rat CD25-PE, clone OX-39 | BioLegend | Cat#202,105 | (1:200) |
| Antibody | (Mouse monoclonal) anti-rat CD8a-PerCP, clone OX-8 | BioLegend | Cat#201,712 | (1:200) |

*Continued on next page*

*Continued*

| Reagent type (species) or resource | Designation | Source or reference | Identifiers | Additional information |
|---|---|---|---|---|
| Antibody | (Mouse monoclonal) anti-rat CD11b/c-PE-Cy7, clone OX-42 | BioLegend | Cat#201,818 | (1:200) |
| Antibody | (Mouse monoclonal) anti-rat CD4-APC-Cy7, clone W3/25 | BioLegend | Cat#201,518 | (1:200) |
| Antibody | (Mouse monoclonal) anti-rat RT1B-AF647, clone OX-6 | BD Biosciences | Cat#562,223 | (1:200) |
| Antibody | (Mouse monoclonal) anti-rat CD45RA-BV421, clone OX-33 | BD Biosciences | Cat#740,043 | (1:200) |
| Antibody | (Mouse monoclonal) Anti-rat CD45-BV605, clone OX-1 | BD Biosciences | Cat#740,515 | (1:200) |
| commercial assay or kit | IMMUNOPREP Reagent System | Beckman Coulter | Cat# 7546999 | Compatible with the COULTER TQ-Prep workstation |
| software, algorithm | Trim Galore | Babraham Bioinformatics, *Krueger et al., 2021* | | https://github.com/FelixKrueger/TrimGalore |
| software, algorithm | CutAdapt | National Bioinformatics Infrastructure Sweden, *Martin, 2022* | | https://github.com/marcelm/cutadapt |
| software, algorithm | Bismark | Babraham Bioinformatics | | https://www.bioinformatics.babraham.ac.uk/projects/bismark/ |
| software, algorithm | MethylKit | Bioconductor | | https://www.bioconductor.org/packages/release/bioc/html/methylKit.html |
| software, algorithm | DMRseq | Bioconductor | | https://www.bioconductor.org/packages/release/bioc/html/dmrseq.html |
| software, algorithm | ClusterProfiler | Bioconductor | | https://bioconductor.org/packages/release/bioc/html/clusterProfiler.html |
| software, algorithm | Homer package | UCSD | | http://homer.ucsd.edu/homer/ |
| software, algorithm | JMP | JMP-SAS | Ver16 | https://www.jmp.com |
| software, algorithm | FlowJo | BD Life Sciences | Ver10.8 | https://www.flowjo.com/ |
| software, algorithm | Cytobank | Beckman-Coulter | | https://www.beckman.com/flow-cytometry/software/cytobank-premium |

## Animals

All experimental procedures were conducted in accordance with the *Guide for the Care and Use of Laboratory Animals* and approved by the NIA(National Institute on Aging) Animal Care and Use Committee (467-CMS-2018 and 469-TGB-2022). Male and female Fischer 344 CDF (F344) rats (were obtained from the NIA Aged Rodent Colony housed at the Charles River Laboratories [Frederick, MD]). The animals were housed with Nylabone supplementation and ad libitum access to food (Envigo 2018åSX) and water. Rats younger than 3 months were housed in groups of three; all other rats were single housed. All rats were maintained on a 12/12 lighting schedule, with all procedures carried out during the light cycle. Rats were habituated to the facility for at least 3 days before sample collection. Some of rats reported here were also analyzed in previous studies (*Levine et al., 2020*; *Finesso et al., 2021*).

## Flow cytometry and sample collection

500 µl of whole blood was collected via retro-orbital bleedings for DNA and FACS analysis. Blood for DNA for DNA methylation was collected in heparinized tubes, spun, and the plasma removed; buffy coat and red blood cells were frozen at −80°C until DNA extraction. Blood for FACS analysis was collected in EDTA-treated tubes, chilled on ice, and 100 µL was stained and then processed using a Beckman Coulter TQ-prep (fixation step) and the Beckman Coulter immunoprep reagent system. For the fresh sample analysis, blood was drawn as above, ACK treated, and immediately stained with

antibodies. For all samples, staining was performed on ice for 30 min in a 100 µl staining cocktail containing the following antibodies at a dilution of 1:200: FITC-conjugated anti-rat CD3 (clone 1F4, Cat#201403), PE-conjugated anti-rat CD25 (clone OX-39, Cat#202105), PerCP-conjugated anti-rat CD8a (clone OX-8, Cat#201712), PE-Cy7 conjugated anti-rat CD11b/c (clone OX-42, Cat#201818), APC-Cy7 conjugated anti-rat CD4 (clone W3/25, Cat#201518) from Biolegend (San Diego, USA), and AF647-conjugated anti-rat RT1B (clone OX-6, Cat#562223), BV421-conjugated anti-rat CD45RA (clone OX-33, Cat#740043), and BV605-conjugated anti-rat CD45 (clone OX-1, Cat#740515) from BD Biosciences (Franklin Lakes, USA). Immunophenotyping data was acquired on a BD FACSCanto II and analyzed using FlowJo (https://www.flowjo.com/). Original FCS files are available on http://flowrepository.org/id/FR-FCM-Z59K.

## DNA methylation analysis

Samples were treated and sequenced for DNA methylation as described in *Levine et al., 2020*. The raw sequencing datasets are available from GEO (GSE161141). To identify differentially methylated loci: we trimmed reads of adapter dimers using Trim Galore (0.4.3) and quality trimmed with a minimum quality score above >25 (--rrbs -q 25). The attached adapter dimers were trimmed using cutadapter. First, bisulfite-converted index (GA and CT conversion) was generated using F344 rat genome with bismark build option and trimmed reads were aligned with bismark (*Krueger and Andrews, 2011*). Once we created aligned reads and corresponding locations, we used the bismark_methylation_extrator tool to summarize the level of methylation in CpG sites (bismark_methylation_extractor -p –comprehensive –no_overlap –bedGraph –counts –buffer_size 16 G [$Aligned read bam file]). Approximately a total 1–2 million sites per sample were predicted with DNA methylated sites (or unmethylated). DMR and blocks of differentially methylated sites were identified with a minimum of 3 CpG sites per block and at least >5% methylation difference (FDR <0.05) using MethylKit (PMID: 23034086 *Akalin et al., 2012*) and DMRseq (*Korthauer et al., 2019*). Functional annotation was performed using ClusterProfiler (*Yu et al., 2012*).

To generate RRBS(Reduced representation bisulfite sequencing) DNA methylation block: RRBS datasets were processed in a uniform way and DNA methylation levels for each sample were extracted. Then, we generated 200 base-pair binned DNA methylation levels across the genome. Each DNA methylation block contains 1–20 CpG sites. We first calculated the average DNAm level per DNAm block for each rat age then used this average DNAm level for mean CpG imputation. We calculated the DNAm level using all methylated and unmethylated sites from the binned CpG sites when the minimum coverage was more than 5. The F344 build of the rat reference genome was used for bisulfite sequencing alignment, and the rn6 genome feature was used to extract genomic annotation information using the Homer package (*Heinz et al., 2010*). Genes, exons, introns, and UTRs were taken from Homer annotation tools (annotatePeaks.pl DMR rn6). TSS-promoter sites were considered the identified DMRs close to TSS (less than 1 kb). TSS-proximal sites were considered the identified DMRs close to TSS (less than 20 kb), and TSS-distal sites were considered the DNAm block whose distance to TSS is above 20 kb. For downstream analysis to find inflection points, we used the TSS-promoter DNAm block for searching inflection point analysis (e.g. Runx1, Dnmt3a).

## Calculation of DNA methylation breakpoints

Genes for DNA methylation breakpoint analysis were selected by first filtering for DMRs that changed as a function of age, determined by a response screening analysis (JMP platform) with an FDR p-value<0.05. The resulting 2,311 sites were then fitted for a linear and a logistic 4 P regression, and the better fit was selected based on Akaike Information Criterion (*Shavlakadze et al., 2019*). Next, we calculated the aging change breakpoint of DMRs that were more likely to be non-linear using the formula in *Figure 6—figure supplement 1*.

## Statistical analysis and software used

Population frequencies were determined with FlowJo v10.8 Software (BD Life Sciences) ViSNE and self-organizing maps analyses were performed using Cytobank (*Kotecha et al., 2010*). All statistical analyses except those specifically detailed otherwise were performed using JMP (JMP, Version 16. SAS Institute Inc, Cary, NC, 1989–2021). The predictive model of age based on leukocyte frequencies was constructed using the least square method with backstep elimination of insignificant variables.

Analysis of pathology prediction was performed using the JMP Decision Tree Platform. Determination of regression formulas and breakpoints was performed in JMP according to the formulas indicated in *Figure 6—figure supplement 1*.

## Acknowledgements

Many thanks to Dr. Rafael de Cabo, Dr. Luigi Ferrucci, Dr. Michel Bernier, and all members of TGB for invaluable support. We would like to thank all the members of the NIA Comparative Medicine Section for their consistent efforts and high standards of animal care. A special thank you to Daniel Ariad for invaluable advice. This research was supported entirely by the Intramural Research Program of the NIH, National institute on Aging.

## Additional information

### Funding

| Funder | Grant reference number | Author |
| --- | --- | --- |
| National Institutes of Health | Intramural NIA | Isabel Beerman |

The funders had no role in study design, data collection and interpretation, or the decision to submit the work for publication.

### Author contributions

Hagai Yanai, Data curation, Formal analysis, Investigation, Methodology, Project administration, Writing – original draft, Writing – review and editing; Christopher Dunn, Conceptualization, Data curation, Formal analysis, Investigation, Methodology; Bongsoo Park, Formal analysis, Methodology; Christopher Coletta, Formal analysis, Investigation, Methodology; Ross A McDevitt, Data curation, Formal analysis, Methodology; Taylor McNeely, Michael Leone, Formal analysis, Investigation; Robert P Wersto, Conceptualization, Resources; Kathy A Perdue, Conceptualization, Data curation, Resources; Isabel Beerman, Formal analysis, Funding acquisition, Investigation, Supervision, Writing – review and editing

### Author ORCIDs

Hagai Yanai http://orcid.org/0000-0003-1742-5411
Christopher Dunn http://orcid.org/0000-0001-7899-0110
Ross A McDevitt http://orcid.org/0000-0003-3722-9047
Isabel Beerman http://orcid.org/0000-0002-7758-8231

### Ethics

All experimental procedures were conducted in accordance with the Guide for the Care and Use of Laboratory Animals and approved by the NIA Animal Care and Use Committee (ASP 467-CMS-2018 and 469-TGB-2022).

### Decision letter and Author response

Decision letter https://doi.org/10.7554/eLife.76808.sa1
Author response https://doi.org/10.7554/eLife.76808.sa2

## Additional files

### Supplementary files

- Transparent reporting form

- Source data 1. Animal pathology report.

- Source data 2. List of rats and detailed peripheral blood leukocyte populations.

- Source data 3. List of differentially methylated regions (DMRs) that are located <20 kb from a transcription start site (TSS).

• Source data 4. Gene Ontology enrichment analysis report.

## Data availability

The data that support the findings of this study are available in the supplementary material and supplementary tables of this article. The DNA methylation raw data is available via GEO Accession (GSE161141). FCS files have been uploaded to FlowRepository (FR-FCM-Z59K). Further information and requests for resources and reagents should be directed to and will be replied to by the corresponding authors.

The following dataset was generated:

| Author(s) | Year | Dataset title | Dataset URL | Database and Identifier |
| --- | --- | --- | --- | --- |
| Yanai H | 2022 | Rat Cross Sectional Aging PB | http://flowrepository.org/id/FR-FCM-Z59K | FlowRepository, FR-FCM-Z59K |

The following previously published dataset was used:

| Author(s) | Year | Dataset title | Dataset URL | Database and Identifier |
| --- | --- | --- | --- | --- |
| Levine M | 2020 | A rat epigenetic clock recapitulates phenotypic aging and co-localizes with heterochromatin-associated histone modifications | https://www.ncbi.nlm.nih.gov/geo/query/acc.cgi?acc=GSE161141 | NCBI Gene Expression Omnibus, GSE161141 |

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
