## [Editor Report]

This paper uses flow cytometry to characterize the changes in immune cell composition of the peripheral blood, as well as DNA methylation, during aging in male rats. Using this data, the authors were able to observe distinct cell composition and DNA methylation profiles with age, providing predictive measures of aging. Additionally, the authors identified a novel marker, CD25+ T cell frequency, as a predictor of age in rats. This resource will be useful to the community as a well-controlled dataset dissecting changes in circulating blood cells with aging in an important mammalian model.

---

## [Decision Letter]

**Decision letter after peer review:**

Thank you for submitting your article "Rat leukocyte population dynamics predicts a window for intervention in aging." for consideration by *eLife*. Your article has been reviewed by 3 peer reviewers, one of whom is a member of our Board of Reviewing Editors, and the evaluation has been overseen by Carlos Isales as the Senior Editor. The reviewers have opted to remain anonymous.

Essential revisions:

Based on our 3 assessments, the most salient points the authors need to address for a revision are:

1. Add additional important context and discussion for choices that may have a significant impact on results (all 3 reviewers):

– Using only males (the expectations if females were included should be discussed);

– Censoring of overtly ill animals (would the results differ if included?), which may bias against frail animals; (If these data points can be included, that would be ideal, if not, the caveat must be discussed explicitly);

– Choice of time breakpoint for the analysis;

– Mentioning the cell composition shifts as caveats for the methylation data.

2. Methodologies need to be reported in more detail and in a more transparent manner for long-term reproducibility (i.e. staining for flow cytometry, availability of the model for other people to reuse, per animal data needs to be reported in a way that allows re-analysis [with identical animal identifiers in all data forms], use of fresh/fixed protocol for the flow cytometry, etc.) (reviewers 1 and 2).

3. Clarifications need to be brought to the relative importance/results linked to myeloid bias (reviewers 1 and 2).

4. If possible, providing more contextualization as compared to studies in other species (reviewer 2).

Finally, we would suggest careful copyediting, including checking the numbers of animals, making sure all figures are referenced and clarifying the study goals in the abstract clarification, may be needed (reviewer 1 and 3).

*Reviewer #1 (Recommendations for the authors):*

1. Since the authors were able to obtain data from 146 animals (Figure 1 data), it would be important to include information on a per animal basis (e.g. average/median number of cells per individual, etc) to have an idea of inter-individual variability.

2. It will be interesting to characterize the age-associated changes with CD4/CD8 T cell ratio in Figure 2 (heatmap), since a decrease in the CD4/CD8 T cell ratio is one of the key characteristics found in human peripheral blood (line 47).

3. Authors should recheck animal numbers in line 76 (146 animals) and line 165 (159 animals). Which number is correct? It will be important to make sure all animals are accounted for.

4. A key characteristic observed in aged peripheral blood is increased myeloid bias. Also, it is described that illness is also associated with increased myeloid bias. Thus, it is slightly puzzling that in Figure 3a more than half of animals with illness (gray dots) present with younger predicted age relative to actual age.

5. Authors discuss possible implications one may infer from the DNA methylation profiles of ALL leukocytes in the peripheral blood (lines 351-359). However, since it is very clear that the cell composition of peripheral blood undergoes significant changes with age, it will be highly relevant to understand the changes in DNA methylation levels along with changes in leukocyte composition. This should be mentioned in the Discussion session.

6. It will be important to describe the staining method used for the flow cytometry (e.g. staining time, antibody concentrations, etc.) should anyone wish to use the protocol from this manuscript.

*Reviewer #2 (Recommendations for the authors):*

1. The authors state: "The second shift point we observed is defined by a marked increase in variance between individual rats which alludes to a pan-hematopoietic loss of homeostasis." This increase in variance can also be explained by the difference in the rate of aging between different individuals. The same goes for the methylation analysis.

2. It is unclear why the authors decided to split the age at 15 months to study the breakpoints of sites. Using this threshold obtained from the cell composition data can introduce a bias into the methylation analysis. The manuscript says: "The methylation switch points defined converged at around 22 months of age and mildly at close to 12 months of age" but it's unclear how these points were obtained. There is no table or figure demonstrating this observation provided in the paper.

3. Genes breakpoint calculations from the data on individual CpGs require a more detailed explanation. Also, are the ApoE and Runx1, mentioned as the genes having a breakpoint at earlier than 15 months, the top hits?

4. The myeloid/lymphoid ratio seems to be a very good predictor by itself. How does it perform in comparison to the proposed model?

5. Description of the differentially methylated regions and the overall change to the methylation landscape would be more insightful if compared to what has been identified in the other species – for example, in another rodent – mouse (Sziráki et al., Aging Cell, 2018).

*Reviewer #3 (Recommendations for the authors):*

1) It is understandable that the authors have not yet completed studies on females, but this should be acknowledged as a major limitation of the work. This should be added to the discussion and the word "male" added to the title and abstract. Also, in the discussion the authors should consider this when they are speaking of similarities and differences between their findings in male rats and comparisons to studies done in humans (potentially in both sexes). As the authors argue that they are looking for a model that is highly translatable to humans, it is important that they also consider translation for both sexes.

2) The authors should completely re-write their abstract. At present is far too vague and speculative and does not give the reader a clear idea of the work they have done.

3) The word fragility is used in the abstract but nowhere else. I suspect they mean frailty, but this should be defined. They should discuss a potential role for frailty in the context of their work in a modified discussion.

4) The authors have not analyzed blood from rats with overt illness. This would have the effect of removing data from the frailest rats from their study. Do the authors have these data to add to the study? This would be very interesting. The absence of these measures should at least be discussed as a limitation in the revised discussion.

---

## [Author Response]

Essential revisions:Based on our 3 assessments, the most salient points the authors need to address for a revision are:1. Add additional important context and discussion for choices that may have a significant impact on results (all 3 reviewers):– Using only males (the expectations if females were included should be discussed);

This study was indeed designed with only males initially to avoid experimental variation at later ages which is a recognized challenge in aging studies. Though we are unable to comprehensively compare female rats at all ages, we too appreciated the relevance of comparing female data. To attempt to address this concern, we analyzed leukocyte frequencies in the peripheral blood of female F344 rats taken from 3 age groups: 5 young (5 months old), 5 middle-aged (15 months old), and 5 old (23 months old) and have included these results (please see Figure 3 —figure supplement 2, Results, and Discussion). To briefly address the finding here, we were pleasantly surprised to see the age prediction model generated from the male rats seems to work robustly on predictions for female data (prediction p <0.0001). This was true even though we were able to observe sex-specific differences, that appear to become exacerbated with age. The blood parameters with sex-specific differences though, do not seem to be dominant parameters driving the prediction model. Though these data would need to be more comprehensively validated, we show both sex-specific aging differences in male and female rats, but also demonstrate the current prediction model robustly predicts the female age as well. We did not have any sick or overtly ill females from this small data set, so we were cautious in the overall conclusions drawn from the additional female samples; however, were still pleased with the data supporting the utility of the male-derived model for female rats as well.

– Censoring of overtly ill animals (would the results differ if included?), which may bias against frail animals; (If these data points can be included, that would be ideal, if not, the caveat must be discussed explicitly);

Our apologies for not making the analyses statement more clear. The excluded animals were part of the originally planned cohort, but these rats did not have blood measurements taken as the veterinarian staff identified them as moribund and the rodents were culled for humane reasons. Their blood was not collected as the rest of the age-matched cohorts were not being assessed at the time of the sick animals’ humane euthanasia. Instead, we attempted to determine how the blood parameters were presented in illness based on veterinary assessment and necropsy on non-overtly ill rodents. We acknowledge that, in retrospect, it would have been beneficial to collect blood from these animals for analysis but unfortunately, we did not. As our study was cross-sectional and blood collections were grouped, we can only get a “snapshot” of the state of the animals during collection. We believe that tracing individual rats in a longitudinal study, ideally using our blood parameters examined in this manuscript, would be an excellent topic to tackle to pre-identify animals that are projected to become moribund.

– Choice of time breakpoint for the analysis;

Please see the specific response to reviewer #2, but briefly: the breakpoints for the analysis were data-driven. In the case of blood parameters, we performed a multiple regression analysis for each cell population and identified time points in which there was a major trajectory shift and most of these shifts appear to converge at approximately 15 months of age. For the methylation data, we performed both linear and logistic regressions on significant age-associated DMRs near TSSs and for each DMR selected the regression with the best fit (based on Akaike Information Criterion). Most sites changed linearly with age, while ~40% showed a better fit to a logistic regression. We used the DMRs that better fit the logistic regression to determine the breakpoint. This was calculated using the breakpoint analysis formula described in Supp Figure 7C. The methylation switch points defined from these analyses, converged at around 22 months of age and mildly at close to 12 months of age.

– Mentioning the cell composition shifts as caveats for the methylation data.

This is an excellent point that we overlooked mentioning and we now raise this caveat in the discussion. As this analysis is a correlative analysis, it is naturally difficult to distinguish between the effects of co-factors that change together. To fully answer this question, we would need to obtain the DNA methylation profiles of individual cell types. This would also require the generation of profiles from each cell type from aged animals, as we hypothesize that specific populations will likely display unique age-associated methylation changes. We have attempted to provide some insight into this question bioinformatically with our existing data by comparing the aging methylation changes to the population frequency changes. While we found some interesting correlations, we do not believe the evidence is strong enough to be included in the main paper and will only be described here.

Briefly, we compared the correlations between the top 2311 DMRs that changed as a function of age with leukocyte population frequencies. To do that, we ran a least-square model that predicts the DNA methylation levels, with or without age as a predictor:

As seen in Author response image 1, Age is a stronger predictor for DNA methylation levels than leukocyte populations on their own. Myeloid/lymphoid ratio is also a strong coefficient, suggesting that cell composition does indeed contribute to the observed DNA methylation changes. However, as seen in Author response image 2, overall R-square values (where the higher the value, the more accurately the parameters describe the observed changes in the DMR) are higher when age is included as a variable than without. This suggests that age, on its own, is a contributor to the DNA methylation changes. Further, the breakpoints between changes in the blood parameters do not overlap directly with the breakpoints in the methylation changes, suggesting the methylation and blood parameter changes may not be completely dependent. While we think this analysis is indicative of a cell composition independent methylation change, we do not believe this evidence is strong enough to make that claim definitively. Therefore, we have added a caveat describing that we cannot exclude the possibility that the methylation changes are driven by changes in the cell composition in the Discussion.

**Author response image 1. sa2fig1:** Heatmap summarizing the impact of each leukocyte population on predicting DNA methylation by least-square model. A least square model was constructed for each DMR of the 2311 top hits using age and leukocyte frequencies as predictive variables. Each row in the heatmap indicates a single DMR. Each column indicates the parameter as indicated. The color corresponds to the P value where blue is < 0.05.

**Author response image 2. sa2fig2:** Summary of R-square values for the models constructed with and without age as a predictor. The R-square values are summarized as a violin plot overlayed with a box blot. Left (blue) are the R-square values with age and right (red) without age.

2. Methodologies need to be reported in more detail and in a more transparent manner for long-term reproducibility (i.e. staining for flow cytometry, availability of the model for other people to reuse, per animal data needs to be reported in a way that allows re-analysis [with identical animal identifiers in all data forms], use of fresh/fixed protocol for the flow cytometry, etc.) (reviewers 1 and 2).

We have expanded the Materials and methods to include all data needed to reproduce the experiment. We have reconfigured the Suppl. Files to be in a more user-friendly format and ensured all identifiers correspond correctly.

3. Clarifications need to be brought to the relative importance/results linked to myeloid bias (reviewers 1 and 2).

Please see specific responses to reviewers. Briefly, we should reiterate that the model was generated on healthy rats, and as such myeloid/lymphoid ratio was not among the strongest predictors. The other parameters (such as CD25+ CD4 cells, NK frequency, CD8^+^ frequency of T-cells, etc.) are the drivers of the healthy model. The myeloid bias has a small dynamic range in the young rats and therefore is not a strong predictor of “healthy” aging. If we had included sick rats in the model, myeloid/lymphoid ratio would indeed be a robust predictor of illness. We believe that a model to predict age should be based on animals that are healthy (as measurable) to reduce the confounding effects of pathologies.

4. If possible, providing more contextualization as compared to studies in other species (reviewer 2).

We were excited to see the analyses when we performed this suggested comparison as we found that the rat blood had quite similar trends to what was seen in human blood with aging, with the dominant changes being loss of methylation in the older cells. Interestingly, the mouse data show similar amounts of both loss and gains of methylation. When we looked for concordance between the three species, we see that the majority of DMRs that gain or lose methylation during aging in rats, also show the same trends in both mouse and human data. Our one large caveat to this analysis is each of these data sets was generated using different platforms with different levels of coverage. While the human global analysis looks similar to what we report in the rats, when examining concordance, there were excessive amounts of missing orthologous data points. We have included these analyses in Figure 4.

Finally, we would suggest careful copyediting, including checking the numbers of animals, making sure all figures are referenced and clarifying the study goals in the abstract clarification, may be needed (reviewer 1 and 3).

We sincerely apologize for the oversights and likely annoyance that they caused. We have done a better job editing and have revised the abstract significantly.

Reviewer #1 (Recommendations for the authors):1. Since the authors were able to obtain data from 146 animals (Figure 1 data), it would be important to include information on a per animal basis (e.g. average/median number of cells per individual, etc) to have an idea of inter-individual variability.

We apologize for the confusion partially due to how we uploaded the Supplemental Tables. Suppl. Table 2 is now presented in a more approachable form and includes all the population frequencies on an individual animal basis to reflect inter-individual variability and allow others to re-analyze the results. As we record a specific number of events in the flow cytometry data collection, the cell number recorded does not accurately reflect interindividual variability on the total cellularity, so we present the data as frequencies. We have also detailed in the Methods that the original flow files are available upon request, and our gating strategy is included in Figure 1 —figure supplement 2.

2. It will be interesting to characterize the age-associated changes with CD4/CD8 T cell ratio in Figure 2 (heatmap), since a decrease in the CD4/CD8 T cell ratio is one of the key characteristics found in human peripheral blood (line 47).

We agree, thank you! We have now included the CD4/CD8 ratio in Figure 4 —figure supplement 1 and discuss it in the results. We do find a modest decline in CD4/CD8 ratio in the first half of life (as seen in humans). Surprisingly, we also observed that in the later stages of aging, there is a strong dysregulation of this ratio. In part, this may be explained by an accumulation of CD25+/CD4^+^ T cells that we described.

3. Authors should recheck animal numbers in line 76 (146 animals) and line 165 (159 animals). Which number is correct? It will be important to make sure all animals are accounted for.

We apologize for this oversight. We have data on a total of 159 male animals. However, we chose to exclude animals younger than 3 months for this study, as we wanted to examine changes in adult hematopoiesis only and not early development, leading to a total number of 146. The numbers are now consistent in the text.

4. A key characteristic observed in aged peripheral blood is increased myeloid bias. Also, it is described that illness is also associated with increased myeloid bias. Thus, it is slightly puzzling that in Figure 3a more than half of animals with illness (gray dots) present with younger predicted age relative to actual age.

This is an excellent point. It is important to stress here that the model was generated using only healthy animals. As such, myeloid/lymphoid ratio was not among the strongest predictors (the top predictors are shown in Figure 3). While the ratio does significantly increase with age, it is not a strong predictor of “healthy” aging. We explored including all rats (healthy and ill) into the prediction model and found that myeloid/lymphoid becomes a significant contributor (p = 0.000232). However, due to the low dynamic range during the early ages, it was not amongst the strongest predictors when excluding sick rats. It seems that on its own, myeloid lymphoid ratio is more predictive of “ailment” than actual age but is a useful predictor of aging and pathology combined.

5. Authors discuss possible implications one may infer from the DNA methylation profiles of ALL leukocytes in the peripheral blood (lines 351-359). However, since it is very clear that the cell composition of peripheral blood undergoes significant changes with age, it will be highly relevant to understand the changes in DNA methylation levels along with changes in leukocyte composition. This should be mentioned in the Discussion session.

We agree. Please see the above response for the related analysis. The discussion now mentions this important caveat.

6. It will be important to describe the staining method used for the flow cytometry (e.g. staining time, antibody concentrations, etc.) should anyone wish to use the protocol from this manuscript.

Done- we have now included timing, volume, and concentration required for the staining. The same staining protocol was used on fixed and fresh blood. Please see the methods section to verify if these modified methods are clear enough for the reproducibility of the protocol.

Reviewer #2 (Recommendations for the authors):1. The authors state: "The second shift point we observed is defined by a marked increase in variance between individual rats which alludes to a pan-hematopoietic loss of homeostasis." This increase in variance can also be explained by the difference in the rate of aging between different individuals. The same goes for the methylation analysis.

That is an excellent point that we overlooked. Thank you. It is now mentioned in the discussion.

2. It is unclear why the authors decided to split the age at 15 months to study the breakpoints of sites. Using this threshold obtained from the cell composition data can introduce a bias into the methylation analysis. The manuscript says: "The methylation switch points defined converged at around 22 months of age and mildly at close to 12 months of age" but it's unclear how these points were obtained. There is no table or figure demonstrating this observation provided in the paper.

We chose the 15-month split based on the shift we saw in the blood population frequencies and indeed it is somewhat arbitrary. However, 16 months of age is also where a drop in the survival curve begins and is also where the male rat reproductive capacity dramatically drops. While we acknowledge that this time point may present a bias, we believe it still accurately reflects a differentiation between early adulthood events and late ones. The convergence of switch points is meant to be descriptive rather than quantitative and is shown in Figure 6a (where the bulges are).

3. Genes breakpoint calculations from the data on individual CpGs require a more detailed explanation. Also, are the ApoE and Runx1, mentioned as the genes having a breakpoint at earlier than 15 months, the top hits?

We have expanded the explanation of breakpoint analysis in an added section of the Materials and methods:

“Genes for DNA methylation breakpoint analysis were selected by first filtering only for DMRs that changed as a function of age, determined by a Response Screening analysis (JMP platform) with an FDR p value < 0.05. The resulting 2,311 sites were then fitted for a linear and a logistic 4P regression and the better fit was selected based on Akaike Information Criterion (Shavlakadze, et al., 19). Next, we calculated the breakpoint on sites that were selected to be less likely to be linear using the formula in Figure 6 —figure supplement 1.”

ApoE and Runx1 were highlighted due to their relevance in the regulation of hematopoiesis. Runx1 is indeed the top hit in the differential methylation analysis and ApoE is more modestly altered.

4. The myeloid/lymphoid ratio seems to be a very good predictor by itself. How does it perform in comparison to the proposed model?

This is indeed an important point brought up by multiple reviewers. As mentioned in the response to reviewer #1, myeloid/lymphoid ratio did correlate strongly with age. When including sick animals, myeloid lymphoid ratio is indeed a very strong predictor of age (p = 0.000232). However, due to the low dynamic range during the early ages, it was not amongst the strongest predictors when excluding sick rats. It seems that on its own, myeloid lymphoid ratio is more predictive of “ailment” than actual age but is a useful predictor of aging and pathology combined.

5. Description of the differentially methylated regions and the overall change to the methylation landscape would be more insightful if compared to what has been identified in the other species – for example, in another rodent – mouse (Sziráki et al., Aging Cell, 2018).

This is an excellent suggestion, thank you. We have performed the comparison between our dataset and publicly available DNA methylation aging datasets from both humans and mice, including the one suggested. We find the results interesting and have expanded main Figure 5 to include these results showing the overall trends towards hypomethylation in the rat and humans, and how mice have a more balanced change- with similar numbers of both gains and loss of methylation associated with aging. However, it is important to note that due to differences in how the DNA methylation data was generated and ultimately sequenced, the comparison is not completely straightforward. We believe that this question deserves a deeper meta-analysis beyond the data we present here, but we feel that analysis may warrant a separate paper with additional data generation required.

Reviewer #3 (Recommendations for the authors):1) It is understandable that the authors have not yet completed studies on females, but this should be acknowledged as a major limitation of the work. This should be added to the discussion and the word "male" added to the title and abstract. Also, in the discussion the authors should consider this when they are speaking of similarities and differences between their findings in male rats and comparisons to studies done in humans (potentially in both sexes). As the authors argue that they are looking for a model that is highly translatable to humans, it is important that they also consider translation for both sexes.

This is an important point, and we have tried to provide a bit of insight into the differences that might be seen in female and male rats. We have edited the title to more explicitly state this paper is describing male rat blood phenotypes. We were pleased to see that the predictive model also robustly predicted the female age albeit in a small sample set. Even given the limited number of female blood donors we could acquire; we saw significant sex-specific blood parameter differences and agree a more in-depth analysis of female rats would be an important follow-up study.

2) The authors should completely re-write their abstract. At present is far too vague and speculative and does not give the reader a clear idea of the work they have done.

We have taken the advice of the reviewer and significantly edited the abstract to more accurately reflect the key findings.

3) The word fragility is used in the abstract but nowhere else. I suspect they mean frailty, but this should be defined. They should discuss a potential role for frailty in the context of their work in a modified discussion.

In the abstract revision we have rephrased the statement that had included fragility. In retrospect, we agree that frailty measurements would have of great interest to measure and include a statement in the discussion about future inclusion of frailty measurements as parameters to determine if such a blood composition analysis could provide insight not only into illness, but frailty as well.

4) The authors have not analyzed blood from rats with overt illness. This would have the effect of removing data from the frailest rats from their study. Do the authors have these data to add to the study? This would be very interesting. The absence of these measures should at least be discussed as a limitation in the revised discussion.

We unfortunately do not have any of the blood parameters from the moribund animals (as described above). We hope to have addressed the major concerns raised and bring up in the discussion how a longitudinal study would be relevant to investigate if the model presented could pre-identify those rats heading towards overt-illness before they became moribund, preventing future studies from losing information by allowing for identification of rodents that will need to be collected / analyzed perhaps before scheduled collection.